# Changes in Combined Lifestyle Risks and the Transition of Activities of Daily Living in the Elderly Population of Taiwan: Evidence from the Taiwan Longitudinal Study on Aging

**DOI:** 10.3390/nu16101499

**Published:** 2024-05-16

**Authors:** Fu-Kuei Chang, Hui-Ting Lin, Jia-Hao Chang, Hsin-Jen Tsai

**Affiliations:** 1Department of Medical Science and Biotechnology, I-Shou University, Kaohsiung 82445, Taiwan; fukuei@isu.edu.tw; 2Department of Health Management, I-Shou University, Kaohsiung 82445, Taiwan; 3Department of Physical Therapy, I-Shou University, Kaohsiung 82445, Taiwan; huitinglin@isu.edu.tw; 4Department of Physical Education and Sport Sciences, National Taiwan Normal University, Taipei City 106209, Taiwan; jhchang@ntnu.edu.tw; 5Department of Nutrition, I-Shou University, Kaohsiung 82445, Taiwan

**Keywords:** activities of daily living, exercising, food consumption, smoking

## Abstract

Functional ability decline occurs with age. This study aims to investigate the associations between the lifestyle factors—exercising, food consumption, and smoking—and the functional ability of the activities of daily living (ADL) by gender. The data were obtained from the Taiwan longitudinal study on aging, a national cohort study. The cross-sectional results demonstrated that the frequency of exercising was negatively associated with ADL in both men and women. Dairy products were positively associated with ADL in men. The longitudinal results illustrated that current and consistent exercising were negatively associated with changes in ADL scores over 4- and 8-year periods in both men and women. Seafood consumption was negatively associated with changes in the subsequent 4-year ADL scores. Past smoking was positively associated with changes in subsequent 4-year ADL scores in men, while current smoking was positively associated with changes in subsequent 8-year ADL scores in women. Therefore, consistent exercising, food consumption, and smoking were associated with ADL functional ability in elderly people, and the associations differed by gender. Elders exercising consistently had good ADL performance and maintained their ADL ability better, especially women. Seafood consumption decreased the risk of ADL decline, while smoking increased the risk of ADL decline.

## 1. Introduction

The elderly population is increasing worldwide and approximately 45% of individuals aged ≥ 60 years are estimated to have difficulty in performing daily activities [1,2]. In Europe, approximately 11–44% of elderly people live with at least one of the activities of daily living (ADL) disability and 8–40% of them live with at least one instrumental ADL (IADL) functional disability [3,4,5]. In Taiwan, the proportion of old adults aged ≥ 65 years with functional disability ranged from 13% in 2010 to 15% in 2015 and is expected to be 16% in 2031 [6,7]. Functional disability in ADL has strong negative effects on independence in elderly individuals and is among the most important attributes leading to institutionalization [8].

Functional decline in ADL in elderly individuals cannot be attributed to a single domain [9,10,11]. Studies have demonstrated that lifestyle factors, including physical activity, diet, and smoking, are associated with functional ability in elderly people and may have positive/negative effects on functional disability [9,10,11]. Longitudinal studies have suggested that regular physical activity is associated with reduced mortality and risk of physical disability [12,13,14]. Several studies have demonstrated the beneficial effects of physical activity programs on functional outcomes in elderly individuals [7,8,9,10]. Diets have also been reported to be related to functional impairments [15,16,17]. An analysis of a 19-year longitudinal study in Japan revealed that a high intake of meat was associated with a decreased risk of subsequent impaired ADL occurrence in Japanese adults aged 47–60 years [15]. Fish and egg intakes were not associated with subsequent impaired ADL occurrence [15]. Less frequent consumption of soy products was associated with the incidence of higher ADL disability for community-living Japanese women aged 75 years and over [18]. A dietary pattern characterized by a relatively low consumption of white rice, but a high consumption of fruits, dairy products, and legumes, was associated with a decreased risk of ADL functional disability in community-dwelling elderly Korean individuals aged ≥ 65 years [16]. A high adherence to a Mediterranean-style diet was cross-sectionally associated with physical performance [19]. A high dietary diversity might have protective effects on the decline of multidimensional outcomes associated with healthy aging, especially physical functions (ADL and IADL) [17].

An aging population with functional disability causes increased strain on healthcare systems and greatly increases public health expenditures [8,20]. Lifestyle factors are modifiable factors, even in old age, and changes in lifestyle factors are likely to have differential impacts and improvements on health [10,11]. Moreover, the impacts could differ by gender [18,21]. In Taiwan, the number of elderly people with functional disabilities continues to increase [6,7]. Few studies have investigated and evaluated the impact of lifestyle factors (physical activity, food consumption, and smoking) and their effect on longitudinal changes in functional ability [14,15,16]. Studies investigating gender differences in these associations are even more limited [18,21]. Therefore, in this study, we aimed to investigate the association between lifestyle factors—exercise, food consumption, and smoking—and ADL functional ability by gender.

## 2. Materials and Methods

### 2.1. Participants and Sampling

The data for this study were sourced from the Taiwan longitudinal study on aging (TLSA), a national cohort study, which was launched in 1989 and seeks to evaluate the health and well-being of elderly individuals in Taiwan. The study design and survey methods of TLSA were documented [22]. TLSA uses a stratified multi-staged equal probability sampling design with the township as the primary sampling unit. The study collected data from 4049 persons aged ≥ 60 years in 1989 and recruited 2462 and 1599 new participants aged 50–66 years in 1996 and 2003, respectively, to replenish the younger population of study participants. Surveys are followed up every three to four years [23,24].

TLSA collected nutritional and dietary data from 1999. Therefore, this study employed data from the 1999 survey as a baseline dataset and data from the 2003 and 2007 surveys as the endpoints. Of the 4440 participants in 1999, 3778 and 3132 successfully completed the interviews in 2003 and 2007, respectively. TLSA was approved by Taiwanese government-appointed representatives. Informed consent was obtained from all participants. This study protocol was approved by the Institutional Review Board of E-Da hospital (03/24/2021). This study was conducted according to the guidelines laid down in the Declaration of Helsinki and all procedures involving human subjects were approved.

### 2.2. Functional Ability

The functional ability of elderly people was evaluated using the six-item ADL and IADL questionnaires. The six-item ADL questions assess whether respondents need help with bathing, dressing/undressing, eating, getting out of bed/standing up/sitting on a chair, moving around the house, and toileting [25]. The response for each item is rated on a 4-point scale ranging from “0”, implying no difficulty, “1”, indicating some difficulties, “2”, suggesting significant difficulties, and “3”, meaning inability to do. The sum of the ADL score was calculated at each wave and was used to measure net changes during the 4-year and 8-year periods.

### 2.3. Exercise Status

Lifestyle factors included exercising, diet, smoking, and alcohol consumption. Exercising habits were based on the following questions: “Do you usually exercise?” The response categories were “no”, “≤2 times/week”, “3–5 times/week”, and “≥6 times/week”. The response options were further divided into two categories: “non-exercise (≤2 times/week)” and “exercise (≥3 times/week).” Moreover, changes in exercise status during the 4- and 8-years periods were examined. A participant who did not exercise between 1999 and 2003 or 1999 and 2007 was classified as “not an exerciser,” while one who exercised habitually in 1999 but not in 2003 or 2007 was classified as a “past” exerciser. Conversely, a participant who did not exercise habitually in 1999 but did so at the time of the study was classified as a “current” exerciser, whereas one with exercising habits between 1999 and 2003 or 1999 and 2007 was classified as a “consistent” exerciser.

### 2.4. Food Frequency

A brief food frequency questionnaire was administered to examine the weekly dietary intake. The weekly consumption frequency of several food categories, including meat/poultry, fish, seafood, eggs, dairy products, beans/legumes, vegetables, fruits, and tea were evaluated. The consumption frequency of each food category was recorded at five levels: never, <1 time/week, 1–2 times/week, 3–5 times/week, and daily.

### 2.5. Smoking and Drinking Status

Smoking status was derived from the following question: “Have you ever smoked?” Responses were categorized as “not a smoker”, “past smoker”, and “current smoker”. Alcohol consumption was derived from the following question: “Do you consume alcohol and how often do you consume alcohol?” Responses were categorized as “no”, “≤2 times/month”, and “≥1 time/week”.

### 2.6. Comorbid Conditions

Comorbid conditions were assessed using a disease list and the total number of diagnosed diseases reported was calculated. Diagnosed diseases included hypertension, diabetes, heart diseases, stroke, cancer, lung diseases, arthritis/rheumatism, gastric ulcer/gastric diseases, liver/gallbladder diseases, hip fracture, cataract, kidney diseases, gout, and bone spurs.

### 2.7. Short Portable Mental State Questionnaire (SPMSQ)

SPMSQ was used to evaluate cognitive function in the TLSA. SPMSQ was employed to assess cognitive function in the elderly [26] and its use was validated in the elderly Taiwanese population [27]. The total SPMSQ score ranged from 0 (worst) to 10 (best). The SPMSQ score at baseline and its change over 4 and 8 years were calculated and used in the analysis.

### 2.8. Analysis

Descriptive data are expressed as proportions or average ± standard deviation. Multivariate linear regression analyses were performed to evaluate the short-term and long-term associations of the frequency of exercising, changes in exercising status, smoking, and the weekly frequency of food consumption with baseline ADL and changes in ADL scores during the 4-and 8-year periods after adjusting for confounding variables. ADL decline was defined as a subsequent 4- and 8-year positive net changes in ADL scores (>0 points), indicating that participants had worse ADL functional ability over 4 and 8 years. Confounding factors included gender, age, years of formal education, alcohol consumption, number of diseases, and baseline SPMSQ scores. Study participants were aged ≥ 65 years and had a baseline ADL/IADL score of ≥0. The SAS software package version 9.2 was used for all statistical analyses. Statistical significance was set at *p* < 0.05.

## 3. Results

Table 1 shows the characteristics of the participants in 1999. Their average ADL, IADL, and SPMSQ scores in 1999 were 1.05, 3.10, and 8.86, respectively. The proportions of study subjects who did not exercise, exercised ≤2, 3–5, and ≥6 times/week were 39.50%, 5.70%, 10.02%, and 44.79%, respectively. The average frequency of food consumption per week for meat/poultry, fish, seafood, eggs, dairy products, beans, vegetables, and fruits was 3.77, 4.44, 1.28, 2.97, 3.56, 3.00, 6.55, and 5.04, respectively. The proportions of smoking, alcohol consumption, and exercise were higher in men than in women (all *p* < 0.0001). The number of diseases, ADL scores, and IADL scores was higher in women than in men (all *p* < 0.01), while the SPMSQ scores were higher in men than in women (*p* < 0.0001). Gender differences were evidenced in the average weekly frequency of meat, fish, seafood, eggs, dairy products, and vegetable consumption (all *p* < 0.001).

Table 2 presents the results of the multivariate linear regression analysis of the cross-sectional association of exercising, smoking, and the weekly frequency of food consumption with ADL scores in 1999. After adjusting for confounding factors, the frequency of exercising ≤2, 3–5, and ≥6 times per week was cross-sectionally and negatively associated with baseline ADL (β = −0.68, −0.77, and −0.82, *p* = 0.0001, <0.0001, <0.0001, respectively).

Moreover, multivariate linear regression analysis stratified by gender showed that the frequency of exercising was cross-sectionally and negatively associated with baseline ADL in both men and women (for men, exercising ≤2, 3–5, and ≥6 times per week, β = −0.57, −0.64, and −0.72, *p* = 0.0121, 0.0004, <0.0001, respectively; for women, β = −0.72, −0.86, −0.89, *p* = 0.0115, <0.0001, <0.0001, respectively). Dairy products were positively associated with baseline ADL in men (β = 0.04, *p* = 0.0174).

Table 3 illustrates the longitudinal associations of exercising, smoking, and the weekly frequency of food consumption with changes in ADL scores during the 4- and 8- year periods. After adjusting for the confounding factors, current and consistent exercising were negatively associated with changes in ADL scores over 4- and 8-year periods (for current exercising, β = −0.85 and −1.14, *p* < 0.0001 and <0.0001, respectively; for consistent exercising, β = −0.96 and −1.05, *p* < 0.0001 and <0.0001, respectively). Past smoking was positively associated with changes in subsequent 4-year ADL scores (β = 0.54, *p* = 0.0087). Seafood consumption was negatively associated with changes in the subsequent 4-year ADL scores (β = −0.07, *p* = 0.0482).

Moreover, analysis stratified by gender demonstrates that current and consistent exercising were negatively associated with changes in ADL scores over 4- and 8-year periods in both men and women (for current exercising, in men, β = −0.52 and −0.71, *p* = 0.0184 and 0.0212, respectively; in women, β = −1.09 and −1.54, *p* < 0.0001 and <0.0001, respectively; for consistent exercising, in men, β = −0.64 and −0.73, *p* = 0.0004 and 0.0048; in women, β = −1.20 and −1.34, *p* < 0.0001 and <0.0001, respectively). Past smoking was positively associated with changes in subsequent 4-year ADL scores in men (β = 0.69, *p* = 0.0004), whereas current smoking was positively associated with changes in subsequent 8-year ADL scores in women (β = 3.51, *p* = 0.0087). Food consumption was not associated with changes in subsequent 4- and 8-year ADL scores in either men or women.

## 4. Discussion

This study demonstrated that the increased frequency of exercising was cross-sectionally and negatively associated with baseline ADL. Current and consistent exercising was longitudinal and negatively associated with changes in ADL scores over 4- and 8- year periods. Elderly current and consistent exercisers were more likely to have better 4-and 8-year ADL ability and maintain their ability longitudinally.

Physical activity has been shown to be related to ADL functional ability and a positive effect of physical activity on ADL functional ability has been observed [9,12,13,14]. By analyzing the data from the CHIANTI study in Italy, Balzi et al. [12] reported that physical activity was associated with a significantly low incidence of ADL disability and the worsening of ADL disability in elderly community-dwelling individuals aged ≥ 65 years. den Ouden et al. [9] reported that physical activity was negatively associated with ADL disability and that it was a significant predictor of ADL disability in middle-aged and elderly persons. Physical activity is a predictor of functional independence, improvement and maintenance in Koreans aged ≥ 65 years in the Korean longitudinal study of aging [14].

Moreover, our study observed that the associations between habitual exercising and longitudinal changes in ADL scores differed by gender. The longitudinal associations of habitual exercising with ADL scores were stronger in women than in men. Elderly women exercising habitually had better ADL ability and capacity to maintain their ability than elderly men.

The present results demonstrated that dairy products and seafood consumption were associated with ADL functional ability. Dairy products were positively associated with baseline ADL in men. Seafood consumption was negatively associated with changes in the subsequent 4-year ADL scores, decreasing the risk of subsequent 4-year ADL decline.

Few studies have investigated the associations between individual food consumption and ADL functional ability, and the related results are inconsistent. Kim and Lee [28] reported that the frequency of dairy consumption was not significantly associated with ADL functional disability in elderly Koreans aged ≥ 65 years. However, Yoshida et al. [29] observed that a high dairy intake was associated with a low risk of ADL functional disability and its progression in the elderly Japanese population aged ≥ 65 years.

Our results also illustrated that the consumption of meat, soybean products, eggs, fruits, and vegetables was not associated with ADL scores and ADL decline. By analyzing a 19-year longitudinal study in Japan, Nakamura et al. [15] found that a higher intake of meat (at least once every 2 days) was associated with a decreased risk of subsequently impaired ADL occurrence in Japanese adults aged 47–60 years. Fish and egg intake was not associated with the subsequent occurrence of impaired ADL [15]. Kim et al. [30] reported that legume and soy product consumption was not significantly associated with ADL disability in Korean elderly people. Kojima et al. [18] observed that less frequent soy product consumption was associated with the incidence of higher ADL disability for community-living Japanese women aged 75 year and over. A cohort study indicated that poor antioxidant and anti-inflammatory food intakes were associated with the odds of developing ADL disability and declining muscle strength in Australian men aged 75 years and over [21]. Comparisons of study results between food consumption and ADL functional ability in countries are difficult due to various habitual dietary behaviors in different countries. Moreover, elderly individuals have a low appetite and decreased food intake when they grow old. The decline in food intake would lower dietary diversity in elderly people [31], which may weaken the association between food consumption and ADL functional ability. A high dietary diversity was suggested to have protective effects in the decline of physical functions in ADL and IADL [17].

This study demonstrated that smoking is not cross-sectionally associated with ADL scores. However, past and current smoking increased the risk of subsequent 4- and 8-year ADL decline in both men and women. Previous studies have reported that smoking is associated with ADL disability in elderly individuals [9,32,33,34]. Parker et al. [32] reported that smoking was associated with ADL limitation in elderly people. den Ouden et al. [9] reported that smoking was positively associated with ADL and a significant predictor of ADL disability in middle-aged and elderly persons in the Netherlands. In the Survey of Health Aging and Retirement in Europe, the onset of smoking increased the risk of ADL functional impairment in community-dwelling elderly individuals, especially among women [33]. Taken together, the experiences of past and current smoking increased the risk of ADL decline in both elderly men and women.

Several limitations should be noted when interpreting the results of this study. Differences in health status and intercurrent illness may affect physical activity/inactivity behavior, and therefore, may be potential confounders in these analyses. Data on physical activity and food consumption were self-reported. Missing values in physical activity, food consumption, or functional ability data may lead to selection bias. Respondents with missing data were more likely to have worse health outcomes or were less health conscious. Objective measures of physical activity in the TLSA survey were absent. The measure of physical activity was coarse and did not evaluate the type or degree of exercise in detail. Comorbidity was assessed by the total number of self-reported diagnosed diseases, and disease severity was not considered. Food frequency data were only collected in a 1999 wave of the TLSA and no recent dietary data were available in the TLSA. Changes developed within the study participants between the two study times (1999–2003 and 1999–2007) and economical/environmental changes also occurred. These fluctuations within participants and in the environment were important and influenced the elderly care. However, these fluctuations were not completely considered in the present study and are one of our limitations.

Our study shows that modifiable lifestyle factors—exercise, food consumption, and smoking—have different impacts on ADL functional ability in older adults. The associations between lifestyle factors and functional ability also differ by gender. Consistently exercising to maintain functional ability in life is encouraged for all old adults, especially for females. For cigarette smokers, smoking cessation is beneficial to prevent the decline of functional ability and should be encouraged for old smokers. Our study results observe the weak associations between individual food consumption and the decline of function ability. The frequency of individual food items may not represent the dietary pattern of old adults. Our future study may focus on the effects of dietary patterns, instead of individual foods, on functional ability. In combination with a recent study [17] and studies in other countries [10,18,21,29], a healthy diverse diet is still encouraged for old people. Our study results suggest that consistent exercise and smoking cessation are important to maintain functional ability in old adults, and that health management strategies for old adults should differ by gender.

## 5. Conclusions

Consistent exercise, food consumption, and smoking were associated with ADL functional ability in elderly people, and this association differed by gender. The longitudinal associations of consistent exercising with ADL were stronger in women than in men. Elderly individuals exercising consistently had good ADL performance and maintained their ADL capacity, especially women. Seafood consumption decreased the risk of subsequent 4-year ADL decline, whereas past and current smoking increased the risk of subsequent 4- and 8-year ADL decline.

## Figures and Tables

**Table 1 nutrients-16-01499-t001:** Characteristics of study participants in 1999 ^1^.

	All	Male	Female	*p*
	*n* = 2986	*n* = 1630	*n* = 1356	
Age (mean ± std) (y)	74.83 ± 6.22	74.65 ± 6.04	75.05 ± 6.42	0.0815
Years of formal education (y)	4.75 ± 4.40	6.49 ± 4.31	2.67 ± 3.51	<0.0001
Smoking (%)				
None	56.97	27.18	92.77	<0.0001
Past smokers	20.40	35.03	2.80	
Current smokers	22.64	37.79	4.42	
Alcohol drinking (%)				
None	78.48	66.32	93.07	<0.0001
≤2 times/month	8.62	12.78	3.61	
≥1 times/week	12.91	20.90	3.32	
Frequency of exercise (times/week) (%)				
None	39.50	34.38	45.65	<0.0001
≤2	5.70	5.34	6.12	
3–5	10.02	8.53	11.80	
≥6	44.79	51.75	36.43	
Number of diseases	2.22 ± 1.75	2.02 ± 1.68	2.47 ± 1.81	<0.0001
ADL scores (mean ± std)	1.05 ± 3.47	0.89 ± 3.26	1.24 ± 3.71	0.0052
IADL scores (mean ± std)	3.10 ± 5.08	2.24 ± 4.56	4.14 ± 5.47	<0.0001
SPMSQ scores (mean ± std)	8.86 ± 1.97	9.47 ± 1.24	8.11 ± 2.40	<0.0001
Average frequency of food consumption per week				
Meat/poultry	3.77 ± 2.58	4.09 ± 2.57	3.39 ± 2.55	<0.0001
Fish	4.44 ± 2.64	4.63 ± 2.55	4.20 ± 2.72	<0.0001
Sea food	1.28 ± 1.61	1.42 ± 1.69	1.11 ± 1.48	<0.0001
Eggs	2.97 ± 2.25	3.25 ± 2.33	2.62 ± 2.11	<0.0001
Dairy products	3.56 ± 3.16	3.33 ± 3.17	3.85 ± 3.13	<0.0001
Beans	3.00 ± 2.35	3.04 ± 2.35	2.96 ± 2.36	0.3700
Vegetables	6.55 ± 1.41	6.47 ± 1.54	6.65 ± 1.23	0.0006
Fruits	5.04 ± 2.52	5.02 ± 2.56	5.08 ± 2.48	0.5182

Std, Standard deviation; ADL, activity of daily living; IADL, independent activity of daily living; SPMSQ, short portable mental status questionnaire. ^1^ Subjects were ≥65 years old and with 0 ≤ baseline ADL scores ≤ 18.

**Table 2 nutrients-16-01499-t002:** Multivariate linear regression analysis of the cross-sectional associations ^1^ of exercise, smoking, and the weekly frequency of food consumption with ADL scores in 1999 ^2^.

	All (*n* = 2523)		Male (*n* = 1394)		Female (*n* = 1129)	
	β	*p*	β	*p*	β	*p*
Frequency of exercise (times/week)						
None	Ref		Ref		Ref	
≤2	−0.68	0.0001	−0.57	0.0121	−0.72	0.0115
3–5	−0.77	<0.0001	−0.64	0.0004	−0.86	<0.0001
≥6	−0.82	<0.0001	−0.72	<0.0001	−0.89	<0.0001
Smoking						
Not a smoker	Ref		Ref		Ref	
Past smokers	0.05	0.6703	0.07	0.5771	0.03	0.9418
Current smokers	−0.13	0.2950	−0.11	0.3661	−0.15	0.6488
Weekly frequency of food consumption						
Meat/poultry	0.01	0.6502	0.03	0.1319	−0.04	0.2335
Fish	0.01	0.4640	0.01	0.7307	0.02	0.4476
Sea food	0.00	0.9159	−0.03	0.2804	0.07	0.1516
Eggs	−0.00	0.9153	−0.00	0.8996	0.01	0.7937
Dairy	0.01	0.2702	0.04	0.0174	−0.01	0.4978
Beans	−0.01	0.6028	−0.02	0.4710	−0.02	0.5986
Vegetables	−0.05	0.1230	−0.05	0.1258	−0.00	0.9904
Fruit	0.02	0.3271	−0.01	0.7845	0.04	0.1509

ADL, activity of daily living; IADL, independent activity of daily living; Ref, reference; SPMSQ, short portable mental status questionnaire. ^1^ Adjusted for gender, age, years of formal education, alcohol drinking, number of diseases and SPMSQ scores at baseline. ^2^ Subjects were ≥65 years old and with 0 ≤ baseline ADL/IADL score ≤ 18.

**Table 3 nutrients-16-01499-t003:** Multivariate linear regression analysis of the longitudinal associations ^1^ of exercise, smoking and the weekly frequency of food consumption with changes in ADL scores during 4- and 8-years periods ^2^.

	ADL Scores (1999–2003)						ADL Scores (1999–2007)					
	All (*n* = 1782)		Male (*n* = 976)		Female (*n* = 806)		All (*n* = 1166)		Male (*n* = 639)		Female (*n* = 527)	
	β	*p*	β	*p*	β	*p*	β	*p*	β	*p*	β	*p*
Exercise status												
Not an exerciser	Ref		Ref		Ref		Ref		Ref		Ref	
Past exercisers	0.15	0.3993	0.43	0.0534	−0.05	0.8646	0.23	0.3518	0.58	0.0663	−0.18	0.6358
Current exercisers	−0.85	<0.0001	−0.52	0.0184	−1.09	<0.0001	−1.14	<0.0001	−0.71	0.0212	−1.54	<0.0001
Consistent exercisers	−0.96	<0.0001	−0.64	0.0004	−1.20	<0.0001	−1.05	<0.0001	−0.73	0.0048	−1.34	<0.0001
Smoking status												
Not a smoker	Ref		Ref		Ref		Ref		Ref		Ref	
Past smokers	0.54	0.0087	0.69	0.0004	−0.51	0.4519	0.06	0.8076	0.14	0.5616	−0.48	0.6429
Current smokers	−0.07	0.8592	−0.04	0.9098	−0.06	0.9406	0.66	0.2281	−0.10	0.8586	3.51	0.0087
Consistent smokers	−0.04	0.8014	0.06	0.6830	−0.84	0.1175	−0.20	0.3962	−0.15	0.5032	−1.02	0.2890
Weekly frequency of food consumption												
Meat/poultry	−0.01	0.7650	−0.02	0.3442	0.02	0.6318	0.01	0.6682	0.00	0.9193	0.03	0.5278
Fish	−0.02	0.3939	−0.01	0.7309	−0.04	0.3282	−0.04	0.1778	−0.00	0.9436	−0.07	0.1478
Sea food	−0.07	0.0482	−0.04	0.3209	−0.10	0.0892	−0.01	0.8448	0.01	0.8640	−0.04	0.6470
Eggs	0.02	0.5062	0.02	0.4790	0.01	0.7723	−0.01	0.8258	0.00	0.9288	−0.03	0.5736
Dairy	0.01	0.7422	0.02	0.3258	−0.01	0.7686	0.00	0.8825	0.00	0.8841	0.00	0.9120
Beans	−0.01	0.6271	−0.00	0.8727	−0.02	0.6600	−0.02	0.5782	−0.04	0.2961	0.00	0.9166
Vegetables	0.03	0.4957	0.01	0.7846	0.06	0.5212	−0.05	0.4690	−0.07	0.2799	0.02	0.9040
Fruit	0.02	0.4785	0.02	0.4008	0.00	0.9475	0.04	0.2827	0.04	0.3505	0.04	0.4740

ADL, activity of daily living; IADL, independent activity of daily living; Ref, reference; SPMSQ, short portable mental status questionnaire. ^1^ Adjusted for gender, age, years of formal education, alcohol drinking status, changes in the number of diseases and SPMSQ scores during 4- and 8-year periods. ^2^ Subjects were ≥65 years old and with 0 ≤ baseline ADL ≤ 18.

## Data Availability

Restrictions apply to the availability of these data. Data were obtained from the Health and Welfare Data Science Center (HWDC), Ministry of Health and Welfare (http://www.mohw.gov.tw/CHT/Ministry/Index.aspx), Taiwan. Due to legal restrictions imposed by the government of Taiwan in relation to the Personal Information Protection Act, data cannot be made publicly available. The use of TLSA data is limited to research purposes only. Requests for data can be sent as a formal proposal to the HWDC, Ministry of Health and Welfare (http://www.mohw.gov.tw/CHT/Ministry/Index.aspx), Taiwan and will be available with the permission of HWDC.

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
