# Peer review of "Changes in Combined Lifestyle Risks and the Transition of Activities of Daily Living in the Elderly Population of Taiwan: Evidence from the Taiwan Longitudinal Study on Aging"

_nutrients, 2024, doi:10.3390/nu16101499_

Round 1

Reviewer 1 Report

Comments and Suggestions for Authors

The manuscript is describing very important details of the correlation of ageing, physical activity and nutritional habits, factors counteracting regression caused by the ageing process. Since it is a large sample findings are very important in the Asian society and health management.

The Introduction and Materials Methods sections are interesting, well written, with enough information.

There are a huge number of data to analyze in this manuscript. Although Authors show all the details of the study in two different time frames (1999-2003 and 1999-2007), changes developing between these times within the sample are not discussed in the Discussion. It might be important in elderly care and in economical/environmental changes as well.

Authors describe the general nutritional habits of the Asian population in concern with ageing, but it would be more interesting to open the scope of discussion, showing other international data. It might be very useful, since in the manuscript authors show many foods (dairy products, eggs, legumes etc.) which are preferred foods in many cultural surroundings.

Although thet time frame of data collection  in the study is ending in 2007, it would be good to include modern data and references showing changes since then in different societies compared to this sample in Taiwan. Some of the literiture, although important, but are really old, please update it!

Author Response

Responses to the reviewer 1

1.     Changes developing within the sample and in economical/environmental changes.

(1)   This limitation was added in the limitation section as suggested and highlighted in green.

2.     International data.

(1)   As suggested, some recent and international studies were added in the introduction and discussion sections and highlighted in green.

3.     Modern data and references 

(1)   As suggested, some recent and international studies were added in the introduction and discussion sections and highlighted in green. The references were updated and highlighted in green.

Reviewer 2 Report

Comments and Suggestions for Authors

The authors in thisstudy have examined the effects of different factors linked to daily activities  and abilities involved daily activities. Following just some considerations and indications:

·      Abstracts: the authors started with the aims of study but it’s useful make just a sentence to introduce.

·      Abstract: please, check the number of words fall within the guidelines of the journal

·      Line 37: there is no more recent data? and the authors state: “….and the number continues to increase” but do not cite the reference, please add it.

·      Lines 39-41: this sentence needs a reference.

·      Line 43: like line 37

·      Lines 57-64: like above, references missed.

·      Line 85: please, indicate the date of ethical approval committee.

·      Line 144: please, use lowercase letter for “p”, check and standardize for entire manuscript

·      Tables: please, use the single line interline caption

·      Tables:the writings are not aligned in the columns, please check

·      Table 2: please, use the same number of decimal points

·      Table 3: look the comment above

·      Line 203: please, report the number of reference just after Balzi et al.[]. Check and standardize for entire manuscript

Comments on the Quality of English Language

Minor editing of English language required

Author Response

Responses to the reviewer 2

1.     Abstracts: a sentence to introduce.

(1)   A sentence was added in the abstract as suggested and highlighted in yellow. 

2.     Abstract: the number of words fall within the guidelines of the journal.

(1)   The guideline of the journal for the word limitation in the abstract are 200. 

(2)   The abstract was revised as suggested and highlighted in yellow. The number of words in the revised abstract is 200.

3.     Line 37: recent data? and do not cite the reference.

(1)   The sentence was revised. The reference was cited in the introduction and highlighted in yellow. 

(2)   The recent studies were added in the introduction section and highlighted in green.

4.     Lines 39-41: this sentence needs a reference.

(1)   The references were cited as suggested in the introduction section and highlighted in yellow.

5.     Line 43: like line 37

(1)   The references were cited as suggested in the introduction section and highlighted in green.

6.     Lines 57-64: like above, references missed.

(1)   The references were cited as suggested in the introduction section and highlighted in yellow.

7.     Line 85: please, indicate the date of ethical approval committee.

(1)   The date of ethical approval was added in the method section and the acknowledge sections (IRB statement) and highlighted in yellow.

8.     Line 144: please, use lowercase letter for “p”, check and standardize for entire manuscript.

(1)   Lowercase letter “p” was used in the results section and the result tables. The revised “p” was highlighted in yellow. 

9.     Tables: please, use the single line interline caption

(1)   The single line interline caption was set in the result tables. The results tables were revised as suggested.

10.  Tables: the writings are not aligned in the columns

(1)   The variables writing was revised as suggested, aligned in the column, and highlighted in yellow.

11.  Table 2: please, use the same number of decimal points

(1)   We used the 2 decimal points for the values in mean, standard deviation and b, and 4 decimal points for the p values.

(2)   The values in the result tables were revised as suggested and highlighted in yellow. 

12.  Table 3: look the comment above

(1)   We used the 2 decimal points for the values in mean, standard deviation and b, and 4 decimal points for the p values.

(2)   The values in the result tables were revised as suggested and highlighted in yellow.

13.  Line 203: please, report the number of reference just after Balzi et al.[]. 

(1)   The citations of the references in the writing were revised as suggested and highlighted in yellow.

14.  Minor editing of English language 

(1)   The edited descriptions and words were highlighted in yellow.

Reviewer 3 Report

Comments and Suggestions for Authors

This study by Fu-Kuei Chang et al. aims to investigate the associations between the lifestyle factors-exercising, food consumption, and smoking and the functional ability of activities of daily living (ADL) by gender. I will then point out possible improvements to the authors.

- I believe that examples of daily activities should be provided, as they may change depending on the society and culture.

- It may also be necessary to point out what is meant by functional impairment.

- The last paragraph could be referenced in the introduction.

- I think the years of data collection would be outdated. Throughout my reading I also wonder how results that are more than 15 years old can be put into practice. If the authors want to defend their study well, they should add practical applications for the present time, including clinical applications.

- It should be considered to include other data from the last 5 or 8 years instead of data from 1999, as 25 years have passed.

- I believe also that much of the references are outdated.

Although the text is well written and the study is correct, I do not believe that these data are of real help to clinical practice with the elderly, even though the results obtained with current data could be similar. I also believe that explanations of relevant concepts are lacking, as well as improvements in aspects such as limitations, future lines and practical applications.

Author Response

Responses to the reviewer 3

1.     Examples of daily activities should be provided, as they may change depending on the society and culture.

(1)   In the TLSA, daily activities were evaluated by ADL questionnaire developed by Katz et al. (1963) and questions included whether respondents need help in bathing, dressing/undressing, eating, getting out of bed/standing up/sitting on a chair, moving around the house, and toileting.

(2)   The descriptions were revised as suggested in method section and highlighted in blue.

2.     It may also be necessary to point out what is meant by functional impairment.

(1)   The word “impairment” may lead to misunderstand. Therefore, the word “impairment” was deleted in the introduction section.

3.     The last paragraph could be referenced in the introduction.

(1)   The references were added in the last paragraph of the introduction section and highlighted in yellow.

4.     Add practical applications for the present time

(1)   As suggested, some recent studies were added in the introduction and discussion sections and highlighted in green.

(2)   As suggested, the descriptions for the future research and the practical applications were added in the end of the discussion section and highlighted in blue.

5.     It should be considered to include other data from the last 5 or 8 years instead of data from 1999, as 25 years have passed.

(1)   Food frequency data was only collected in 1999 wave of the TLSA and no recent dietary data was available in the TLSA. This is one of the limitations in our study. This limitation is added and described in the limitation section and highlighted in blue.

(2)   As suggested, some recent studies were added in the introduction and discussion sections and highlighted in green.

6.     The references are outdated.

(1)      As suggested, some recent studies were added in the introduction and discussion sections and highlighted in green. The references were updated.

7.     Explanations of relevant concepts are lacking, as well as improvements in aspects such as limitations, future lines and practical applications.

(1)   The limitations of this study were revised as suggested and highlighted in blue and green.

(2)   As suggested, some recent studies were added in the introduction and discussion sections and highlighted in green.

(3)   As suggested, the descriptions for the future research and the practical applications were added in the end of the discussion section and highlighted in blue.

Round 2

Reviewer 3 Report

Comments and Suggestions for Authors

I appreciate the authors for implementing some of the requested modifications. However, I believe it is crucial to update the data, despite acknowledging that it is a limitation of the study. This restriction has significant importance, and therefore, it is essential for the authors to evaluate whether any therapeutic intervention can truly be based on this data. I am sorry but I still consider that it should not be published.